# Preventive Effect of the Japanese Traditional Herbal Medicine Boiogito on Posttraumatic Osteoarthritis in Rats

**DOI:** 10.3390/medicines7120074

**Published:** 2020-12-04

**Authors:** Jun Oike, Takayuki Okumo, Hideshi Ikemoto, Yusuke Kunieda, Shingo Nakai, Haruka Takemura, Hiroshi Takagi, Koji Kanzaki, Masataka Sunagawa

**Affiliations:** 1Department of Physiology, School of Medicine, Showa University, 1-5-8 Hatanodai, Shinagawa-ku, Tokyo 142-8555, Japan; jun0724@med.showa-u.ac.jp (J.O.); h_ikemoto@med.showa-u.ac.jp (H.I.); situation2@med.showa-u.ac.jp (Y.K.); yawatuki@icloud.com (H.T.); suna@med.showa-u.ac.jp (M.S.); 2Department of Orthopedic Surgery, Showa University Fujigaoka Hospital, 1-30 Fujigaoka, Aoba-ku, Yokohama City, Kanagawa 227-8501, Japan; htakagi@med.showa-u.ac.jp (H.T.); kkanzaki@med.showa-u.ac.jp (K.K.); 3Department of Judo Seifuku and Health Sciences, Faculty of Health Promotional Sciences, Tokoha University, 1230 Miyakoda-cho, Kita-ku, Hamamatsu City, Shizuoka 431-2102, Japan; nakai4n5@gmail.com

**Keywords:** boiogito, knee osteoarthritis, rat

## Abstract

**Background:** Considering the anti-inflammatory properties of the Japanese traditional Kampo medicine Boiogito (BO), we aimed to investigate the therapeutic effect of BO to prevent the development of knee osteoarthritis (KOA) in rats with surgically induced KOA. **Methods:** Destabilization of the medial meniscus (DMM) was performed to induce osteoarthritis in the right knees of 12-week-old Wistar rats under general anesthesia. The rats were orally administered 3% BO in standard powder chow for 4 weeks after surgery (controls: *n* = 6; sham group: *n* = 6; DMM group: *n* = 5; DMM + BO group: *n* = 5). During this period, the rotarod test was performed to monitor locomotive function. After 4 weeks, histological assessment was performed on the right knee. **Results:** Oral administration of BO improved locomotive function in the rotarod test. Walking time on postoperative days 1, 14, or later was significantly longer in the DMM + BO group than in the DMM group. Histologically, the DMM group showed significant progression of KOA, which, in the DMM + BO group, was strongly suppressed, as assessed by the Osteoarthritis Research Society International score. **Conclusions:** Our results showed that oral administration of BO had a clinically preventive effect on early stage posttraumatic KOA.

## 1. Introduction

The number of osteoarthritis (OA) patients is >300 million worldwide. OA can affect any joint of the body, such as the hip, spine, hand, and especially, the knee [1]. Knee osteoarthritis (KOA) is characterized by the gradual progression of functional disorders, such as restricted range of motion due to degenerative changes in the knee joints, often with pain and swelling, and KOA impairs the daily life activities of patients [2,3]. Moreover, the pathophysiology of KOA is characterized not only by abnormal articular cartilage but also by synovitis, osteophyte synthesis, excessive turnover of subchondral bone, and periarticular soft tissue contracture, which all lead to malfunction of the affected joint [4]. Although the etiology of KOA is not fully understood, obesity, age, metabolic syndrome, and knee injury are considered to be risk factors [5,6], and posttraumatic KOA accounts for 12% of all cases. Knee injury is apparently the main factor for development of KOA in younger people [7].

Currently, KOA can be treated surgically and non-surgically. In non-surgical treatments, oral or topical administration of non-steroidal anti-inflammatory drugs (NSAIDs) or oral selective cyclooxygenase-2 (COX-2) inhibitors, intra-articular injection of hyaluronic acid, land-based exercise, dietary weight management with physical exercise, and mind–body exercise (such as Tai Chi and Yoga) have been recommended as effective strategies for KOA patients [8]. Surgical treatments, such as total/uni-compartmental knee arthroplasty (or osteotomy around the knee), are generally considered when non-surgical approaches have failed to control several clinical symptoms of KOA. Since posttraumatic joint instability due to meniscal tear or anterior cruciate ligament (ACL) rupture can cause destruction of the articular cartilage [9], orthopedic surgery aims to achieve meniscal repair or ACL reconstruction. These operative procedures can delay KOA progression; however, they cannot completely prevent the devastation of articular cartilage, even after ACL reconstruction [10]. In fact, KOA degenerative changes tend to be worse when they coexist with meniscal tear due to ACL rupture [11]. Although various kinds of biochemical mediators, such as interleukin-1β (IL-1β), tumor necrotic factor-α (TNF-α), nitric oxide (NO), and proteolytic enzymes, reportedly can have important roles in the progression of posttraumatic KOA [9], there are no effective preventive treatments for the pathological changes caused by KOA.

Several compelling reports and a review article on pharmaceutical treatment for KOA by using herbs or flavonoids have been published [12,13,14]. Some plant extracts, such as rosehip and curcumin, have shown therapeutic roles for OA by suppressing inflammatory mediators, including IL-1β, TNF-α, and NO; proteolytic enzymes, such as matrix metalloproteases; and/or a disintegrin and metalloproteinase with thrombospondin motifs. Choi et al. [15] reported the preventive effect of SKI 306X, a mixture of three herbs, against collagenase-induced arthritis in rabbits. Interestingly, SKI 306X potentially can inhibit proteoglycan degradation, unlike dexamethasone or NSAIDs.

Boiogito (BO), a traditional Japanese herbal medicine (Kampo) composed of six medicinal plants, may also be a potent medication for preventing osteoarthritis. BO is effective for the symptoms of chronic fatigue or hyperhidrosis, as well as leg edema or painful arthritis, and its use for these indications has been approved by the Japanese Ministry of Health, Labour, and Welfare. Some clinical studies have suggested that BO potentially can alleviate inflammation and hydrarthrosis in KOA [16,17]. Majima et al. [16] studied the effects of BO on KOA and joint effusion. Oral administration of BO improved functional capacity while stair climbing, and significantly reduced joint effusion without severe adverse effects. Although BO could have some therapeutic effects for KOA, there is no evidence that BO suppresses KOA progression, especially when evaluating locomotive dysfunction and structural destruction, such as articular cartilage devastation. If a preventive effect against KOA can be demonstrated, BO may become a therapeutic option for early stage KOA. The study’s aim was to evaluate the disease modifying effect of BO on osteoarthritis in a rat model of surgically induced KOA.

## 2. Materials and Methods

### 2.1. Animals

Male Wistar rats at 12 weeks old, average weight 300–350 g, were purchased from Nippon Bio-Supp. Center (Tokyo, Japan). All rats were fed standard powdered rodent chow (CE-2; CLEA Japan, Tokyo, Japan) and water ad libitum before and after the surgical procedure. The animals were housed two to three per cage in an animal room with a controlled environment (12-h light/dark cycle, temperature 20–25 °C, and humidity 50–60%). The experimental protocols (Figure 1) were approved by the Institutional Ethics Committee for Care and Use of Animals of Showa University (certificate number: 09056, date of approval: 1 April 2019).

### 2.2. Drug Administration

The dry powdered extract of BO was supplied by Tsumura & Co. (TJ-20; Tokyo, Japan) and contained a dry extract of the mixed drug substances consisting of *Sinomenium* stem 5.0 g, *Astragalus* root 5.0 g, *Atractylodes lancea* rhizome 3.0 g, Jujube 3.0 g, *Glycyrrhiza* 1.5 g, and Ginger 1.0 g. These herbs were mixed and extracted with purified water at 95.1 °C for 1 h, and the soluble extract was then separated from the insoluble residue and dried by removing water under reduced pressure. The dry powdered extract of BO was mixed with powdered chow at a concentration of 3% and fed to the BO-treated rats. The rats not treated with BO were fed powdered chow only. The concentration of BO was chosen on the basis of the effective doses recommended by a previous report [18].

### 2.3. Surgery

Destabilization of the medial meniscus (DMM) was adopted as a KOA-inducing model [19]. We divided rats into four groups: control, sham, DMM, and DMM + BO-treated groups. DMM and sham surgeries were performed on the right knee fixed in a flexed position under isoflurane (Fujifilm Wako Pure Chemical Corp., Osaka, Japan) inhalation general anesthesia. A skin incision was made in the midline of the right knee to expose the quadriceps muscle and patellar tendon. The medial edges of the patella tendon and medial joint capsule were separated (Figure 2A), and then, the patella was dislocated from the lateral femoral condyle (Figure 2B). The medial meniscotibial ligament (MMTL) was then able to be observed. The MMTL was transected, and the medial meniscotibial joint capsule was horizontally cut in the DMM and DMM + BO groups (Figure 2C) to induce meniscal extrusion out of the bone contact area between the femur and tibia (Figure 2D). No cutting procedure was performed in the sham group. Finally, the medial joint capsule and patella tendon were sutured with 6-0 Vicryl^®^ (Ethicon Inc., Somerville, NJ, USA), and the subcutaneous layer was sutured with 5-0 Vicryl^®^.

### 2.4. Meniscus Extrusion Ratio

To assess the validity of the DMM operation, the degree of extrusion of the medial meniscus was evaluated. The animals were intraperitoneally anesthetized with pentobarbital sodium (50 μg/kg; Somnopentyl, Kyoritsu Seiyaku, Tokyo, Japan) and intracardially perfused with phosphate-buffered saline at pH 7.4 until all the blood had been removed from the system. After perfusion with 4% paraformaldehyde in 0.1 M phosphate-buffered saline, only the right leg was amputated because the contralateral side of the knee joint may not yet have an osteoarthritic change [20]. Then, the right knee joint was fixed with 4% paraformaldehyde for 3 days and decalcified with a 20% EDTA solution for 21 days. Then, the knee joint was cut in a coronal shape amid and along the medial collateral ligament fiber, and the degree of meniscal lateral deviation out of the outer edge of the femoral condyle was measured by using a Stemi 305 stereomicroscope (Carl Zeiss, Oberkochen, Germany) (Figure 3A). This measurement was performed three times in each sample, and the average of these three was finally defined as the meniscus extrusion (ME) ratio, which is the length of the extruded meniscus/the length of the medial meniscus × 100 (%) [21] (Figure 3B). Specimens with <20% of the ME ratio in the DMM and DMM + BO groups were excluded from this study because they were considered to indicate failure of the DMM surgery.

### 2.5. Rotarod Test

The rotarod test easily validates the effects of drugs, brain disorders, and disease on rodent motor coordination and fatigue tolerance [22,23]. The test was performed by a third person who was not engaged in the surgery and did not know the grouping. The influences on locomotive performance were assessed prior to and 1, 3, 7, 14, 21, and 28 days after the DMM surgery by using an automated accelerating rotarod apparatus (LE8305), with a lane width of 75 mm and rod diameter of 60 mm (Panlab Harvard Apparatus, Barcelona, Spain) (Figure 1). The rats were forced to make forward walking movements to circumvent falling. The rat was trained 5 min per day for 2 days to stay on the drum. The rotarod was accelerated 5–40 rpm over 30 s. The time that rats remained on the rotarod was recorded. All values were averaged over three consecutive measurements. The cutoff time was set at 45 s.

### 2.6. Histological Analysis

After measuring the ME ratio with a stereomicroscope, decalcified knees were embedded in paraffin, and specimen preparation and tissue staining were performed following the recommendations of the Osteoarthritis Research Society International (OARSI) [24]. All sections were sliced 4-μm thick every 200 μm from the center of the medial collateral ligament. The sections were visualized by using an Olympus BX 53 microscope (Olympus, Tokyo, Japan) after toluidine blue staining was performed. At least three tissue sections were prepared from each specimen. Histological assessment was determined following the OARSI scoring system [24]. The slides were evaluated by two independent histologists. Cartilage degeneration was evaluated on a scale of 0–15 points, subchondral bone destruction was evaluated on a scale of 0–5 points, osteophyte formation was evaluated on a scale of 0–4 points, and the total score ranged from 0–24 points, in which a lower score indicated less joint degeneration. The average value of the three slices with the poorest scores was taken as the OARSI score of each knee.

### 2.7. Statistical Analysis

Data are represented as the mean ± SD of multiple repeats of the same experiment for the data of ME ratio and rotarod test, and median and interquartile range for the histological analysis (control, *n* = 6; sham, *n* = 6; DMM, *n* = 5; DMM + BO, *n* = 5). Statistical analysis was performed by using one-way analysis of variance and the Tukey–Kramer method in JMP^®^ Pro version 14.0 software (SAS Inc., Cary, NC, USA). *p* values < 0.05 were taken as indicating statistically significant differences.

## 3. Results

### 3.1. Meniscus Extrusion Ratio

To assess the validity of the DMM operation, the degree of extrusion of the medial meniscus was evaluated 4 weeks after the operation (Figure 3A,B). The ME ratios were 10.2 ± 2.3% in the control group, 11.4 ± 3.0% in the sham group, 42.2 ± 7.3% in the DMM group, and 37.8 ± 5.8% in the DMM + BO group (Figure 3C). The differences were not significant between the control and sham groups. However, the ME ratios were significantly higher in the DMM and DMM + BO groups than in the control and sham groups, but the differences between the DMM and DMM + BO groups were not significant. These findings strongly suggested that meniscal extrusion and displacement of the medial meniscus out of the femoral condyle occurred with DMM surgery.

### 3.2. Rotarod Test

Rotarod performances were assessed prior to and 1, 3, 7, 14, 21, and 28 days after the DMM surgery (Figure 4). In the acute phase (on days 1 and 3), although walking time on the rotarod apparatus was slightly improved in the control group, there was no significant difference. The latencies to fall off the rotarod apparatus (walking time) were significantly lower in the DMM group than in the control group (*p* < 0.01). However, the decreases were inhibited in the DMM + BO group, especially significant on day 1 (*p* < 0.05). In the chronic phase (on days 14, 21, and 28), the latencies were significantly lower in the DMM group than in the control and sham groups (*p* < 0.01) Those decreases were significantly inhibited in the DMM + BO group (on day 14; *p* < 0.01, on days 21 and 28; *p* < 0.05). Furthermore, walking time on day 28 in the DMM + BO group had no significant difference than that in the control group.

### 3.3. Histological Analysis

Histological assessment was determined by OARSI score 4 weeks after the operation [23]. Although cartilage degeneration was significantly greater in the DMM group than in the control and sham groups, the degeneration observed in the DMM group was alleviated in the DMM + BO group (Figure 5A). The OARSI cartilage scores were 0.0 points (0.0–0.25) in the control group, 0.33 points (0.08–0.58) in the sham group, 2.67 points (2.25–3.00) in the DMM group, and 0.67 points (0.33–2.33) in the DMM + BO group. There was no significant difference in the OARSI cartilage scores between the control and sham groups. The OARSI cartilage score was significantly higher in the DMM group than in the control and sham groups (*p* < 0.01) but was significantly lower in the DMM + BO group than in the DMM group (*p* < 0.05) (Figure 5B). The total OARSI scores, including the cartilage score, osteophyte formation score, and subchondral bone damage score, were 0.0 points (0.0–0.25) in the control group, 0.33 points (0.08–0.58) in the sham group, 5.00 points (3.00–5.25) in the DMM group, and 0.67 points (0.33–4.33) in the DMM + BO group. As in the cartilage score results, there were no significant differences between the control, sham, and DMM + BO groups; however, the total OARSI score was significantly higher in the DMM group than in the other groups (vs. control and sham; *p* < 0.01, vs. DMM + BO; *p* < 0.05) (Figure 5C).

## 4. Discussion

Although the pathological mechanism of KOA has not been fully elucidated, KOA is a disease involving articular cartilage but also synovium, subchondral bone, and periarticular soft tissue [25]. Inflammation of synovium is associated with alterations in the adjacent cartilage. Catabolic and pro-inflammatory mediators, such as cytokines, NO, and prostaglandin E2, produced by the inflamed synovial membrane, lead to excessive production of proteolytic enzymes associated with cartilage degradation [26]. Current strategies for treatment of KOA are aimed at relieving clinical symptoms, such as pain or swelling of the knee joint and disability of walking, and delaying the progression of KOA, such as cartilage degradation, osteophyte synthesis, or subchondral bone destruction [27,28].

Considering the latency to fall off the rotating drum in the rotarod test in this study, administration of BO improved walking ability in the DMM rat model. Although the rotarod test can detect general locomotive function in experimental rodent models [29], it can also reflect, at least in part, pain-related locomotive dysfunction derived from surgical invasion, especially within a few days after surgery. In the present study, walking time on the rotarod apparatus was improved by administration of BO not only on days 14–28 (chronic phase) but also a few days after the operation (acute phase) (Figure 4). Among the crude drugs of BO, *Astragalus* root [30], *Atractylodes lancea* rhizome [31], Jujube [32], *Glycyrrhiza* [33], and Ginger [34] have been reported to have analgesic actions in various kinds of animal models of pain. BO can potentially relieve acute postoperative pain.

Meniscus extrusion is one of the factors for progression for KOA. In humans, medial meniscus posterior root tear reportedly contributes to meniscus extrusion, possibly because of extensive mechanical loading of the articular cartilage and subchondral bone [35]. The DMM rat was established to develop KOA by transection of the MMTL, which induces medial meniscus extrusion. We confirmed that the anterior part of the medial meniscus was completely dislocated from the femoral condyle in rats with DMM surgery (Figure 3). In the present study, the patellar tendon and medial joint capsule were split in the sham group, but KOA did not occur as evaluated by OARSI score. Surgical invasion did not evidently induce osteoarthritic change in the knee joint. Therefore, it is considered that the biomechanical disorder caused by meniscus extrusion strongly influences the development of KOA in the DMM group. Surprisingly, BO inhibited progressive destructive damage of the knee joint, although meniscus extrusion was observed as well in the DMM group. To the best of our knowledge, this is the first report in which BO has been shown to have a preventive effect on posttraumatic KOA.

There has been some basic research suggesting that BO could have a therapeutic effect against KOA, especially on joint fluid retention and the inflammatory response in KOA. Fujitsuka et al. [36] showed that BO inhibited IL-1β secretion due to synovitis in an ACL-transected rat KOA model. Indomethacin, an NSAID, also has been shown to decrease IL-1β in the synovial fluid but failed to inhibit joint fluid retention in the rats’ knee joints. These results are similar to those in the clinical study mentioned above [16]. Takenaga et al. [37] reported a suppressive effect of BO extract on MMP-13 production in rats with adjuvant-induced arthritis as well as an inhibitory effect on the secretion of IL-1β and MMP-13 in cultured chondrocytes. These findings suggest that BO could inhibit the secretion of pro-inflammatory cytokines, decrease excessive joint effusion, and stimulate proper irrigation of the knee joint with joint fluid. Further study is needed to elaborate on the morphopathogenetic and therapeutic mechanism of BO in the DMM-induced KOA model to prove the link between tissue and clinical changes in the rats.

As mentioned above, BO is composed of six crude drugs and contains various kinds of chemical components, but it is not clear which components contribute to the preventive effect against KOA progression. Previous studies have investigated some potential components for alleviating the development of KOA. *Sinomenium* stem, a principal crude drug of BO, has been shown to have anti-inflammatory activity [38]. In animal models of adjuvant and collagen-induced arthritis, *Sinomenium* stem improved symptoms and decreased the expressions of pro-inflammatory cytokines, such as IL-1β and TNF-α [39]. *Astragalus* root also has been shown to have anti-inflammatory activity by decreasing the pro-inflammatory cytokine, TNF-α, although in a rat autoimmune myocarditis model and mouse diabetic model [40,41]. Given those findings, a synergistic effect by combining the anti-inflammatory activity of *Sinomenium* stem and *Astragalus* root may provide the characteristic therapeutic effect of BO against early stage KOA. This ideal combination of chemical components in BO can be applied clinically, such as in the perioperative period for meniscal tears and ACL ruptures, to improve excessive joint effusion and prevent the progression of KOA. Further investigation is needed.

Several limitations of our study need to be considered. First, as mentioned above, we have to elucidate the therapeutic mechanism to prove drug efficacy. We are performing subsequent studies such as immunohistochemical staining and the gene expression of proteins related to cartilage metabolism, including collagen type II and matrix metalloproteinase (MMP)-13. However, the results from this study would be a proposal of a new strategy for posttraumatic KOA. Second, in this study, we estimated the effect for 4 weeks after surgery to assess a preventive effect on the early stage of posttraumatic KOA. However, we need to verify the efficacy longitudinally. When the disease state lasts long, the contralateral side must also be influenced [20]. Therefore, in the future, we will follow the long-term changes of both knees.

## 5. Conclusions

In conclusion, oral administration of BO was found to have some therapeutic effect on preventing clinical osteoarthritic changes in a posttraumatic KOA-inducing rat model. BO potentially could be applied in various clinical situations, such as for perioperative administration to relieve pain, and prevent the future progression of KOA.

## Figures and Tables

**Figure 1 medicines-07-00074-f001:**
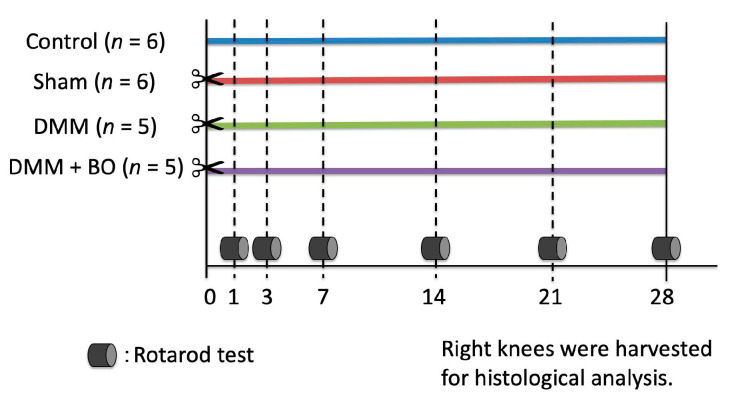
Experimental protocol of the present study. Destabilization of the medial meniscus (DMM) and the sham surgery were performed on the right knee on day 0 (✂). Boiogito (BO) was mixed in the chow at a concentration of 3%. The rotarod test was performed prior to and 1, 3, 7, 14, 21, and 28 days after the DMM surgery. The rats were sacrificed, and their right knees were harvested for histological analysis on day 28.

**Figure 2 medicines-07-00074-f002:**
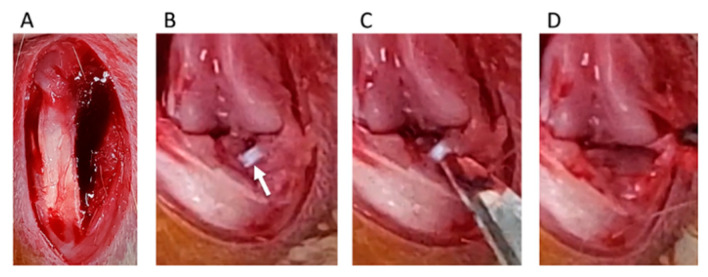
Surgical procedure of destabilization of the medial meniscus (DMM). (**A**) The medial edge of the patella tendon and the medial joint capsule are separated. (**B**) The medial meniscotibial ligament (MMTL) is detected. White arrow indicates the MMTL. (**C**) The MMTL is transected, and the medial meniscotibial joint capsule is horizontally cut. (**D**) Meniscal extrusion is confirmed.

**Figure 3 medicines-07-00074-f003:**
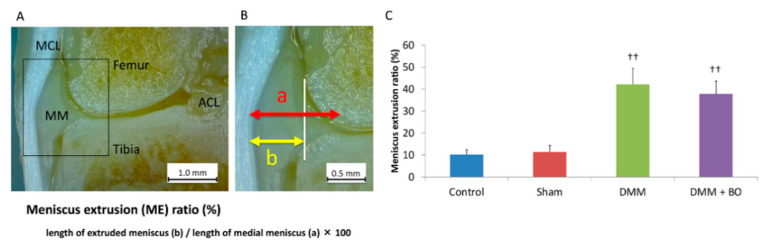
Meniscus extrusion (ME) ratio. (**A**) Coronal view of the medial tibiofemoral compartment. The medial collateral ligament and tibial footprint of the anterior cruciate ligament is confirmed. MCL—medial collateral ligament; ACL—anterior cruciate ligament; MM—medial meniscus. (**B**) Definition of ME ratio. White vertical line indicates the medial edge of the femoral condyle. (**C**) ME ratio; quantitation of the extruded meniscus. Bars show the mean ± SD (*n* = 6 in the control and sham groups, *n* = 5 in the DMM and DMM + BO groups). The stars indicate a significant difference from the control group by the Tukey–Kramer test (^††^
*p* < 0.01 vs. Control).

**Figure 4 medicines-07-00074-f004:**
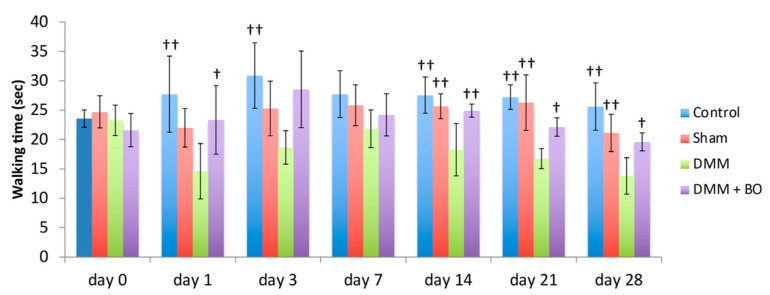
Locomotive functional test with the rotarod test. The stars indicate a significant difference from the DMM group by the Tukey–Kramer test (^†^
*p* < 0.05, ^††^
*p* < 0.01 vs. DMM).

**Figure 5 medicines-07-00074-f005:**
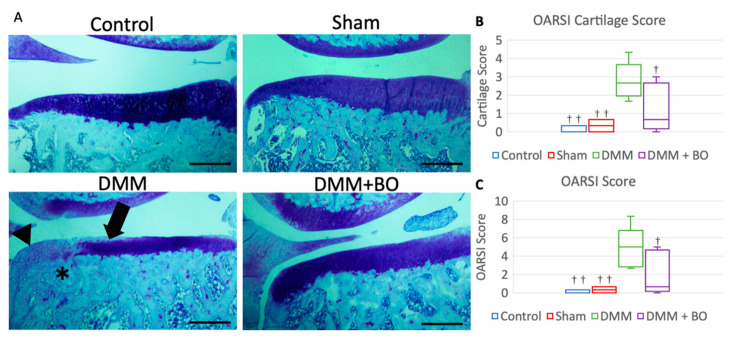
Histological analysis of the medial tibial cartilage. (**A**) Representative images of coronal sections of the medial tibial plateau stained with toluidine blue. Magnification: * 40. Scale bars = 500 μm Arrow (➡) indicates degeneration of the extracellular matrix in articular cartilage. Asterisk (*) indicates subchondral bone damage. Arrowhead (▶) indicates osteophyte synthesis. (**B**) OARSI cartilage score. (**C**) Total OARSI score. Boxplots denote median values and interquartile ranges. Vertical bars show ranges (*n* = 6 in the control and sham groups, *n* = 5 in the DMM and DMM + BO groups). The marks indicate a significant difference from the DMM group by the Tukey–Kramer test (^†^
*p* < 0.05, ^††^
*p* < 0.01 vs. DMM).

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
