# Peer review of "Preventive Effect of the Japanese Traditional Herbal Medicine Boiogito on Posttraumatic Osteoarthritis in Rats"

_medicines, 2020, doi:10.3390/medicines7120074_

Round 1

Reviewer 1 Report

The authors of the manuscript have tried to examine effects of the Japanese traditional Kampo medicine (Boiogito, BO) on the symptoms of experimentally-induced rat model of osteoarthritis. The compound (just likely to be one of anti-inflammatory stuffs) exhibited prophylactic effects on the OA symptoms, however, the manuscript needs fundamental revisions for acceptance in Medicines.

Major concerns
1. DMM + vehicle group should be set to confirm the net effect of BO.
2. Dose-dependent effects of BO should be provided.
3. OARSI (cartilage) score should not be represented by mean +/- SD. Instead, use median with interquartile range.
4. Behavioral tests and histological observations should be tested in a blinded fashion.

Minor concerns
1. Consider the appropriate decimal points, for example, in line 170, “10.2% ± 2.26%” should be “10.2 ± 2.3%”.

Author Response

I would like to thank you for your review. I have revised my manuscript as follows.

Major concerns

  1. DMM + vehicle group should be set to confirm the net effect of BO.

> We used the pure extract of the herbal medicine BO without diluting agent and mixed it into a powdered chow. So, think that the DMM group is equivalent to DMM + vehicle group.

  1. Dose-dependent effects of BO should be provided.

> Thank you for pointing it up. Unfortunately, we don’t have some other meaningful data of the dose-dependent effect of BO. Actually, we started the experiment with a concentration of 5% of BO as a pilot study. As mentioned in the Materials and Methods, the concentration of BO was chosen on the basis of the effective doses recommended by a previous report (Shimada et al. Evid Based Complement Alternat Med 2011, 2011, 931073.). In their study, 1% and 3% were investigated, and 3% was recommended. So, we switched to 3% BO rather than 5% BO that we didn’t get enough number of the sample from the 5% BO rats. We are studying more with more or lesser concentration of BO, like 0.5%, 1%, or 5%.

  1. OARSI (cartilage) score should not be represented by mean +/- SD. Instead, use median with interquartile range.

> Thank you for pointing it out. We revised it (p.6 L209~).

  1. Behavioral tests and histological observations should be tested in a blinded fashion.

> Just as you said, these experiments had been performed by a third person who is not engaged in the process of surgery or staining. We added the mention of it (p.4 L149 and p.5 L165).

Minor concerns

  1. Consider the appropriate decimal points, for example, in line 170, “10.2% ± 2.26%” should be “10.2 ± 2.3%”.

> Thank you for pointing it out. We revised it.

Reviewer 2 Report

Manuscript relates to quite "dangerous" and controversial topic - the alternative medicine! However, it also might represents some interesting issues for part of specialists in the clinical medicine!

Despite this, I have some minor and some very serious objections for this manuscript:

Introduction - OK;

Materials and methods: 1) sham animal group is OK, but the first objection relates to the relatively healthy leg of other side in the same animal, - here tissue usually also show the changes, although they are compensatory ones. But this info can also be important and unfortunately is absent in this case! 2) absolutely important objection relates to the evaluation of degenerative changes in the affected knee; nowadays the staining with only with toluidine blue doesnt work for the tissue quality evaluation. The degeneration/regeneration processes have to be evaluated by help of immunohistochemistry at least: different MMPs, TIMPs, their ration, IL1, cellular activity indicators, apoptosis. Not less important are tissue quality markers: OPN, OC, collagen type II. If this is not done, the results are not acceptable and believable! 3) the minor remark, - way of euthanasia of animals also should be described;

Results: morphology pictures are not valid al all, as they represents small magnification, are repainted and unconvincing. Magnifications are not indicated, too.

Discussion: the main objection relates to the absence of discussion regarding the confounding factors because they are not described and justified here, but could play an important role for the healing process. It is very difficult to prove the connection between the oral administration of BO and tissue changes in the improvement of knee osteoarthritis. Also the role of surgery (traumatic process) is not discussed, but the trauma might be very strong factor for tissue changes...

Conclusions: superficial and blurry, should be shorten seriously and develop in more precise way.

References: Ok (however, one "old" reference is mentioned here.

Author Response

I would like to thank you for your review. I have revised my manuscript as follows.

Introduction - OK;

Materials and methods:

1) sham animal group is OK, but the first objection relates to the relatively healthy leg of other side in the same animal, - here tissue usually also show the changes, although they are compensatory ones. But this info can also be important and unfortunately is absent in this case!

> Sorry, we didn’t observe it, because 4 weeks after surgery an appreciable change in the contralateral side is not observed according to the previous report (*). In the present study, we investigated the changes for 4 weeks after surgery to assess a preventive effect on early stage post-traumatic KOA. However, as you said, from a long-term view, the contralateral side must also be influenced, so, we will have to follow the long-term change of both sides in the future. We added the comment (p.4 L136, p.9 L323).

(*) Gardiner MD, et al. Transcriptional analysis of micro-dissected articular cartilage in post-traumatic murine osteoarthritis. Osteoarthritis Cartilage. 2015;23(4):616-28.

2) absolutely important objection relates to the evaluation of degenerative changes in the affected knee; nowadays the staining with only with toluidine blue doesn’t work for the tissue quality evaluation. The degeneration/regeneration processes have to be evaluated by help of immunohistochemistry at least: different MMPs, TIMPs, their ration, IL1, cellular activity indicators, apoptosis. Not less important are tissue quality markers: OPN, OC, collagen type II. If this is not done, the results are not acceptable and believable!

> We plan to study some associated factors in the near future. Immunohistology is going to be undergone with MMP-13 and type 2 collagen to qualify the articular cartilage, and we are going to do TRAP/ALP staining for verifying the abnormal bone turnover in the subchondral bone. Furthermore, we are planning the in vitro study for chondrocyte and osteoclast to elucidate the direct biological effect of Bo to those cells. We are now studying and modifying the protocol of those investigations. (P.9 L317)

3) the minor remark, - way of euthanasia of animals also should be described.

> Thank you for pointing it out. We added ways of anesthesia and euthanasia (P. 4 L132).

Results: morphology pictures are not valid al all, as they represent small magnification, are repainted and unconvincing. Magnifications are not indicated, too.

> We changed the size of pictures and added a magnification ratio and scale bars (p.7).

Discussion: the main objection relates to the absence of discussion regarding the confounding factors because they are not described and justified here but could play an important role for the healing process. It is very difficult to prove the connection between the oral administration of BO and tissue changes in the improvement of knee osteoarthritis. Also, the role of surgery (traumatic process) is not discussed, but the trauma might be very strong factor for tissue changes.

> We revised and appended our opinions (p.8).

Conclusions: superficial and blurry, should be shorten seriously and develop in more precise way.

> Also revised (p.9). Thank you for pointing it out.

References: Ok (however, one "old" reference is mentioned here).

Reviewer 3 Report

This paper studied the effect of Boiogito on posttraumatic osteoarthritis in rats.

Comments

  1. Figure 2 is already published exactly the same on BMC Sports Sci Med Rehabil 4, 3 (2012). This figure can be deleted or placed in supplement.
  2. Please mention the comparative groups with statistical difference in Figure 5. The authors may need to provide comments on the increase of walking time in the control group just after feeding between day 0 and day 1, and the difference between control and DMM + BO at day 28.  

Author Response

I would like to thank you for your review. I have revised my manuscript as follows.

Figure 2 is already published exactly the same on BMC Sports Sci Med Rehabil 4, 3 (2012). This figure can be deleted or placed in supplement.

> Thank you for pointing it out. We deleted it.

Please mention the comparative groups with statistical difference in Figure 5.

> We changed the explanation (p.6 L197).

The authors may need to provide comments on the increase of walking time in the control group just after feeding between day 0 and day 1, and the difference between control and DMM + BO at day 28.

> Thank you for pointing it up. We provided the comments in the manuscript (p.6).

Round 2

Reviewer 2 Report

Dear Authors,

You have made some changes in the manuscript, but still I have two minor and one major objection:

minor:

1) line 165-166 is not correctly written. You noted, that "This assessment was also carried out by a third person who is not engaged in the staining process", but for histological evaluation usually we use the following "The slides were evaluated by two independent histologists (or histopathologists)". There is no need to involve the third person, but the main point is 0 two specialists and independent ones...

2) Conclusions. They are much better now, but you havent done research neither on the morphopathogenetical mechanisms, nor on the deep morphology on the tissue, thus you cant prove the link between the pathogenesis and improvement, and thus please, speculate only on the clinical improvement, for instancy: "...oral administration of BO was found to have some therapeutic effect on preventing clinical osteoarthritic changes in a posttraumatic KOA-inducing rat model..." This would be believable and acceptable from the scientific point of view! Correct this also in the Abstract!

3) My main objection is still the Discussion! I would kindly suggest you to remove the text starting from the Line 284-288 (from "in the present study".... until the ..."knee joint"), as these are ONLY SPECULATIONS, and you cant prove them! Continue with the all paragraph starting from the line 290, but instead of "Further investigation is needed"put an other sentence "Further study is needed to elaborate on the morphopathogenetic and therapeutic mechanism of BO in the DMM-induced KOA model to prove the link between tissue and clinical changes in the rats" (something like that)...

In this case your discussion will look more objective as the superficial changes (suppressing degeneration of articular cartilage, subchondral bone damage, and osteophyte formation) mentioned by you without the detection of modern tissue factors are not valid for the prove of real tissue improvement! Or you can even say this, that you will check in future the connection between the very routine tissue changes like what you found now and fine molecular tissue degeneration/regeneration/remodelation factors... Then you will be on the "safe side" and nobody could blame you for overspeculations etc.

Author Response

Reviewer 2 Evaluation

I would like to thank you for your review. I have revised my manuscript as follows.

1) line 165-166 is not correctly written. You noted, that "This assessment was also carried out by a third person who is not engaged in the staining process", but for histological evaluation usually we use the following "The slides were evaluated by two independent histologists (or histopathologists)". There is no need to involve the third person, but the main point is 0 two specialists and independent ones...

Thank you for your suggestion. We revised the sentence where you pointed out.

2) Conclusions. They are much better now, but you haven’t done research neither on the morphopathogenetical mechanisms, nor on the deep morphology on the tissue, thus you can’t prove the link between the pathogenesis and improvement, and thus please, speculate only on the clinical improvement, for instancy: "...oral administration of BO was found to have some therapeutic effect on preventing clinical osteoarthritic changes in a posttraumatic KOA-inducing rat model..." This would be believable and acceptable from the scientific point of view! Correct this also in the Abstract!

Thank you for suggesting the minor change in the manuscript. We revised it.

3) My main objection is still the Discussion! I would kindly suggest you to remove the text starting from the Line 284-288 (from "in the present study".... until the ..."knee joint"), as these are ONLY SPECULATIONS, and you can’t prove them! Continue with the all paragraph starting from the line 290, but instead of "Further investigation is needed"put an other sentence "Further study is needed to elaborate on the morphopathogenetic and therapeutic mechanism of BO in the DMM-induced KOA model to prove the link between tissue and clinical changes in the rats" (something like that)...

In this case your discussion will look more objective as the superficial changes (suppressing degeneration of articular cartilage, subchondral bone damage, and osteophyte formation) mentioned by you without the detection of modern tissue factors are not valid for the prove of real tissue improvement! Or you can even say this, that you will check in future the connection between the very routine tissue changes like what you found now and fine molecular tissue degeneration/regeneration/remodelation factors... Then you will be on the "safe side" and nobody could blame you for overspeculations etc.

Thank you for pointing it up. We changed the sentence in the page 7.

Round 3

Reviewer 2 Report

Thank you, the manuscript is improved and can be published.